# Small Molecule Anion Carriers Correct Abnormal Airway Surface Liquid Properties in Cystic Fibrosis Airway Epithelia

**DOI:** 10.3390/ijms21041488

**Published:** 2020-02-21

**Authors:** Ambra Gianotti, Valeria Capurro, Livia Delpiano, Marcin Mielczarek, María García-Valverde, Israel Carreira-Barral, Alessandra Ludovico, Michele Fiore, Debora Baroni, Oscar Moran, Roberto Quesada, Emanuela Caci

**Affiliations:** 1UOC Genetica Medica, IRCSS Istituto Giannina Gaslini, 16145 Genova, Italy; ambrygianotti@hotmail.com (A.G.); valeriacapurro@yahoo.it (V.C.); livia.delpiano@gmail.com (L.D.); 2Departamento de Química, Facultad de Ciencias, Universidad de Burgos, 09001 Burgos, Spain; marcin.mielczarek@poczta.onet.pl (M.M.); magaval@ubu.es (M.G.-V.); icarreira@ubu.es (I.C.-B.); 3Istituto di Biofisica, Consiglio Nazionale Delle Ricerche (CNR), 16149 Genova, Italy; ale.ludo89@gmail.com (A.L.); fiore@ge.ibf.cnr.it (M.F.); dbaroni@ge.ibf.cnr.it (D.B.); oscar.moran@cnr.it (O.M.)

**Keywords:** cystic fibrosis, anionophore, bronchial epithelial cells culture, ion transport, periciliar mucus properties

## Abstract

Cystic fibrosis (CF) is a genetic disease characterized by the lack of cystic fibrosis transmembrane conductance regulator (CFTR) protein expressed in epithelial cells. The resulting defective chloride and bicarbonate secretion and imbalance of the transepithelial homeostasis lead to abnormal airway surface liquid (ASL) composition and properties. The reduced ASL volume impairs ciliary beating with the consequent accumulation of sticky mucus. This situation prevents the normal mucociliary clearance, favouring the survival and proliferation of bacteria and contributing to the genesis of CF lung disease. Here, we have explored the potential of small molecules capable of facilitating the transmembrane transport of chloride and bicarbonate in order to replace the defective transport activity elicited by CFTR in CF airway epithelia. Primary human bronchial epithelial cells obtained from CF and non-CF patients were differentiated into a mucociliated epithelia in order to assess the effects of our compounds on some key properties of ASL. The treatment of these functional models with non-toxic doses of the synthetic anionophores improved the periciliary fluid composition, reducing the fluid re-absorption, correcting the ASL pH and reducing the viscosity of the mucus, thus representing promising drug candidates for CF therapy.

## 1. Introduction

Transepithelial ions and water transport, which are essential for the proper functioning of epithelial tissues, are controlled by different proteins distributed non-symmetrically to the two sides of the epithelium [1]. The airway surface liquid (ASL) is important for the maintenance of innate defence mechanisms that protect the airway epithelium from inhaled pathogens and other noxious agents [2]. The ASL is composed of mucus and a periciliary liquid layer (PLC) whose integrated activities are required for mucociliary clearance [3]. The thickness and ion concentrations of ASL are carefully maintained through the chloride and bicarbonate secretion, predominantly via the cystic fibrosis transmembrane conductance regulator (CFTR) protein, and the sodium absorption mediated by the epithelial sodium channel (ENaC) [4].

Cystic fibrosis (CF) is a genetic disease caused by mutations in the CFTR gene. This results in the lack of a functional CFTR protein expressed in epithelial cells and thus defective chloride and bicarbonate secretion affects the upper and lower airway, intestine, endocrine and reproductive organs [5,6]. Loss of CFTR protein function imbalances the transepithelial homeostasis, which as a direct consequence generates a reduced and more acidic ASL. This in turn impairs the beat of the cilia ending in an accumulation of sticky mucus in the affected area [7,8]. This environment inhibits the antimicrobial activity favouring the survival and proliferation of bacteria and contributing to the genesis of CF lung disease [9,10].

Ionophores are small molecules capable of facilitating the transmembrane transport of ions [11]. These molecules can be used as synthetic tools to manipulate cellular homeostasis and their application in ion-channel replacement therapies has already been proposed [12]. Thus, substitution of the faulty CFTR chloride and bicarbonate transport activity by anion selective ionophores, named anionophores, represents an attractive therapeutic application for these molecules [13].

The outcome of current CF approved treatments is far from optimal and depends dramatically on the type of mutation harboured by the patient [14,15]. Providing an alternative transport pathway to CFTR, anionophores may correct the defective activity independently from the genetic mutations expressed by the CF patient. This is especially relevant in the case of patients with non-sense mutations lacking the CFTR protein, who currently do not have any therapeutic options [16]. Encouragingly, Amphotericin B, a channel-forming molecule capable of facilitating unselective ion permeation, has been recently shown to restore ASL pH, viscosity and antibacterial activity in cultured airway epithelia derived from genetically diverse humans with CF [17].

Previous studies have identified several anionophores inspired by natural products such as prodiginines and tambjamines, able to transport halides in mammalian cells at non-toxic concentrations [18,19,20,21]. Transmembrane transport activities comparable to that of the native CFTR were measured for the best tambjamines candidates [22]. Although promising, these results were not obtained in a physiologically relevant disease model.

In this regard, we have now tested these compounds in cultured epithelia derived from human bronchial epithelial cells (hereafter termed HBEC). HBEC are considered an adequate model due to their ability to maintain most of the morphological and functional characteristics of the airway epithelium in vivo [23,24].

Since CF is characterized by a reduced ASL volume and by the presence of a sticky and viscous mucus [25], the aim of this work was to assess the impact of two synthetic anionophores (named MM3 and MM34) on the periciliary properties of the CF airway epithelium. Here, we have demonstrated that these synthetic anionophores were able to correct mucus viscosity, ASL pH and fluid transport to values comparable to non-CF epithelia and therefore they may represent a promising starting point for the development of new drug candidates for CF therapy.

## 2. Results

### 2.1. Anionophore-Driven Anion Transport in Large Unilamellar Vesicles (LUVs)

Transmembrane anion transport activity of compounds MM3 and MM34 (Figure 1a) was first assessed in model 1-palmitoyl-2-oleoyl-glycero-3-phosphocholine (POPC) LUVs. The normalized chloride efflux elicited by different concentrations of MM3 and MM34 (0.025 to 5 µM) is shown in Figure 1b,c. Representation of the normalized chloride external concentration at 300 s against the anionophore concentration, leads to a curve that was fitted using Hill analysis. Calculated EC_50_ for MM34 is 130 ± 7 nM, whereas for MM3 it is even lower at 35 ± 4 nM (Figure 1d). The fact that these values were found in the nanomolar range confirms the extraordinary anionophoric activity displayed by these compounds, capable of facilitating the exchange of chloride and bicarbonate across model phospholipid membranes very efficiently [20].

### 2.2. Halide Transport Experiments on Fisher Rat Thyroid Cells

Before assaying the anionophores in a complex system as the HBEC monolayers, we checked whether MM3 and MM34 were able to transport halides in a simpler cell model. For this purpose, we used a Fisher Rat Thyroid (FRT) cell line transfected with the iodide-sensitive YFP. The anionophore-facilitated halide transport was then expressed as the initial quenching rate of the Yellow Fluorescent Protein (YFP) upon an extracellular application of iodide. We already demonstrated that MM3 facilitates a significant iodide transport in FRT cells, requiring 2.8 ± 0.6 µM to induce half of the maximum transport rate [22]. Instead, the halide transport capacity of MM34 was not previously assayed in mammalian cells. In Figure 2a we present the time course of the quenching of the iodide-sensitive YFP when FRT cells were treated with different concentrations of MM34. The concentration to elicit half of the maximum transport for MM34 is 2.1 ± 0.6 µM, similar to the value observed for MM3. However, the maximum quenching rate (mQR) of MM3 (334.9 ± 35.8 s^−1^) is 3.75-fold greater than the mQR measured for MM34 (89.4 ± 10.6 s^−1^). The dose-response curve for both anionophores is presented in Figure 2b.

### 2.3. Toxicity of the Anionophores

Before moving to primary bronchial epithelial cells experiments, we assessed the effect of the compounds on HBEC viability. Trypan blue survival test yielded *TD*_50_ values of 0.90 ± 0.14 µM and 1.33 ± 0.2 µM for MM3 and MM34, respectively. However, this apparently low TD_50_ values were mitigated by a small maximum toxicity, *Tox_max_*, 28.1 ± 2.0% (71.9% survival at 4 µM, the highest anionophore concentration tested) for MM3 and *Tox_max_*, 42.0 ± 2.8% (58% survival at 4 µM) for MM34. Since we were seeking for an anionophore concentration that could represent a 95% cell survival for the length of the experiments, we decided to continue the analysis with a 0.25 µM dose for both compounds.

### 2.4. Transepithelial Conductance

We evaluated the effect of the anionophores on the transepithelial electrical conductance (1/resistance) using differentiated non-CF and CF epithelia carrying the F508del homozygous mutation. After measuring the basal transepithelial resistance, cell monolayers were treated for 24 h with 0.25 µM of the anionophores or 0.1% vehicle (DMSO); the transepithelial resistance was then measured at the end of the treatment (Table 1). The converted conductance values, recorded before and after the incubation, are plotted in Figure 3. The treatment with MM3 and MM34 significantly increased transepithelial conductance of epithelia, while it remained invariant in the control (Table 2). We concluded that the anionophores may promote a transmembrane ion transport in CF epithelia due to a transcellular pathway.

### 2.5. Impact of the Anionophores on the ASL Properties

To evaluate the alterations of the ASL properties, induced by the treatment with the anionophores, a volume of buffer containing 0.25 μM of the anionophores was applied to the apical surface of CF epithelium. For control, 0.1% of the vehicle DMSO was applied to the apical surface of the CF and non-CF epithelia. We have previously demonstrated that this concentration does not affect the vitality and function of differentiated epithelia [26]. Indeed, we have not observed differences in any in any of the analysed parameters between non-DMSO (data not shown) and DMSO treatment.

After 24 h the remaining apical fluid and mucus mixture was recovered to evaluate the fluid re-absorption rate as well as the pH and the micro-rheological properties of the ASL mucus. The fluid re-absorption rate was measured gravimetrically after 24 h, as shown in Figure 4. Control CF epithelia has a re-absorption rate of 2.29 ± 0.09 μL/h/cm^2^ (*n* = 7), that is significantly higher than the re-absorption rate of the non-CF epithelia (1.67 ± 0.06 μL/h/cm^2^, *n*= 9). When CF epithelia were treated with MM3 or MM34, re-absorption was significantly reduced to values of 1.75 ± 0.13 μL/h/cm^2^ (*n*= 7) and 1.92 ± 0.08 μL/h/cm^2^ (*n* = 7), respectively. Notice that these values are not significantly different to those measured in the non-CF epithelia.

The pH measured in the ASL of non-CF epithelia was 7.35 ± 0.05 (*n* = 8), while a more acidic ASL was found in CF-epithelia (6.70 ± 0.03, *n* = 17). These data are in agreement with those reported by others, reflecting the efficacy of our protocol [27]. Interestingly, the ASL pH of CF-epithelia treated with the anionophores was partially restored, yielding 7.01 ± 0.04 (*n* = 9) upon incubation with MM3, and 6.95 ± 0.02 (*n* = 9) in epithelia treated with MM34 (Figure 5).

Micro-rheology measurements showed that the viscosity of the ASL was significantly larger on CF-epithelium (140.47 ± 20.54 cPoise, *n* = 8), compared with the non-CF epithelium control (59.50 ± 5.30 cPoise, *n* = 6). Conversely, we observed a significant reduction of the ASL mucus viscosity in CF-epithelia treated with MM3 (61.90 ± 5.56 cPoise, *n* = 7) or MM34 (77.74 ± 12.14 cPoise, *n* = 8), achieving values comparable to those of non-CF epithelia (Figure 6).

Summarizing, these findings suggest that the treatment of F508del-CFTR epithelia with the anionophores decreases the transepithelial fluid transport, increasing airway hydration, counteracts the ASL acidification and reduces the mucus viscosity retrieving ASL properties similar to those of the non-CF epithelia.

## 3. Discussion

Traditionally, the CF therapeutic strategies were aimed at ameliorating the debilitating symptoms of the disease, such as the lung bacterial colonization with recurrent infections, difficulties in breathing and digestive disorders. All these strategies are relevant to improve the patient’s quality and expectance of life, but do not correct the underlying basic defect of CF [28].

Important advances toward the pharmacological correction of the CFTR activity have been made in the last few years. The development of new therapeutic drugs, named potentiators and correctors, addressing the molecular basis of CF represents a milestone in the route to provide therapeutic options for these patients. On the other hand, the variability in the clinical response of patients to these new treatments [29,30] and the existence of a cohort of CF patients lacking the CFTR protein (class 1 mutations for instance) underscores the urgent need of new therapies aimed at developing alternative anion transport pathways independent of CFTR.

In this regard, we have targeted the development of small molecules able to transport anions selectively across lipid bilayers as a genetic agnostic therapeutic approach for CF. We, and others, had previously identified synthetic anionophores capable of promoting anion transport in mammalian cells [31]. Nevertheless, to the best of our knowledge, no demonstration of the usefulness of these compounds in relevant disease models has been reported.

For this purpose, we decided to explore the impact of two selected hit compounds from our drug discovery program, MM3 and MM34, in the properties of the ASL of HBEC epithelia cultures. These two compounds are inspired by the structure of tambjamine natural products.

We first demonstrated their excellent anionophoric properties in phospholipid vesicles model. Our experiments showed the ability of these compounds to efficiently exchange bicarbonate and chloride across the model lipid bilayers. Their ability to promote anion transport was subsequently demonstrated in FRT cells as mammalian cell model. A significant concern to take into account was the potential toxicity derived from the impaired homeostasis provoked by the action of an external agent facilitating anion permeation. Toxicity studies in non-CF HBEC allowed us to determine 0.25 µM as a safe concentration inducing no significant toxicity, a requisite to undergo their study in a more physiological epithelial model. Homozygous F508del-CFTR HBEC epithelia as relevant CF model and non-CF HBEC monolayers as control, were employed in our studies. F508del is the most frequent and better characterized mutation in the CF patient population and causes a severe phenotype of the disease.

Transepithelial electrical conductance measurements indicated a significant increase of the conductance after application of both MM3 and MM34. This result is in agreement with an augmented anion permeability induced by these compounds. We then sought to investigate the potential impact of this enhanced anion permeability on some key features of ASL such as fluid re-absorption, pH and mucus micro-rheology. Impaired homeostasis in CF-epithelia results in an increase of the fluid re-absorption leading to a reduction of the ASL volume. Our data is in agreement with these observations, and the fluid re-absorption rate of untreated CF epithelia was found approximately two fold higher than that in non-CF epithelia. Application of both MM3 and MM34 reduced the re-absorption rate to values comparable to that of non-CF epithelia (Figure 4).

ASL is characteristically more acidic in cultures of CF airway epithelia [32]. This is related to a reduced bicarbonate secretion and causes an incorrect formation of the mucin net with inadequate rheological properties for the normal airway physiology [33]. Actually, we observed these differences in the ASL pH of our primary human airway cultures. The fact that the anionophores were able to facilitate the transport of bicarbonate prompted us to investigate the effect of these compounds in the ASL pH. Treatment with MM3 and MM34 resulted in a significant increase of the ASL pH up to close to 0.3 pH units (Figure 5). This increase in the pH has been linked to a better environment for the innate defence in these epithelia and has been demonstrated to enhance host defences in CFTR-null pigs in vivo [24,34].

Finally, we assessed the impact of the treatment of the epithelia with the anionophores in the viscosity of the secreted mucus. It has been demonstrated that the pharmacological correction of F508del-CFTR with the CF drug lumacaftor improves significantly mucus viscous-elastic properties, but not the fluid re-absorption [35]. This is, perhaps, correlated with the increase of bicarbonate permeability of the VX809-corrected CFTR [36]. Our data confirmed that MM3 and MM34 are able to reduce the mucus viscosity in CF epithelia to an extent comparable to non-CF HBEC epithelia (Figure 6). This decrease in viscosity, along with a better hydration of the ASL, would improve the mucus clearance with beneficial effects in the antibacterial activity, as explained before.

In conclusion, we have demonstrated that the tested small anionophore molecules, MM3 and MM34, correct significantly abnormal ASL parameters in vitro in HBEC epithelia cultures from CF patients. These parameters are intimately linked to the pathophysiology of the CF pulmonary disease. Therefore, these data constitute a promising proof of concept for the usefulness of small molecule anionophores capable of facilitating chloride and bicarbonate transport, in order to develop future CF drugs independent of the CFTR function and thus of the genotype of the CF patient.

## 4. Materials and Methods

### 4.1. Synthesis of the Anionophores

The chemical structures of MM3 and MM34 are displayed in Figure 1a. Compound MM3 was synthesized as previously reported [20].

Compound MM34 was prepared by reaction of 5-(1-(4-chlorophenyl)-1*H*-1,2,3-triazol-4-yl)-3-methoxy-1*H*-pyrrole-2-carbaldehyde (253 mg, 0.84 mmol) with cyclohexylamine (191 µL, 1.67 mmol, 2.0 equiv.) in chloroform (10 mL), employing acetic acid as catalyst (50 µL). The resulting solution was stirred at 60 °C for 5.5 h. Upon cooling to room temperature, the chloroform was evaporated under reduced pressure and the residue re-dissolved in dichloromethane (30 mL). This solution was washed with a 1 M HCl aqueous solution (3 × 20 mL), dried over anhydrous sodium sulphate, filtered and concentrated to dryness. The residue was recrystallized from a mixture of dichloromethane and n-hexane to give compound MM34 as a brown solid (yield: 100 mg, 29%). The compound was fully characterized by NMR and MS (the spectra can be found in Appendix A). The chemicals employed in the synthesis of the compound were purchased from Sigma-Aldrich (St. Louis, MO, USA).

### 4.2. Chloride Efflux in Large Unilamellar Vesicles (LUVs)

The transmembrane anion transport activity of the compounds was tested in model 1-palmitoyl-2-oleoyl-*sn*-glycero-3-phosphocoline (POPC) large unilamellar vesicles (LUVs), with a mean diameter of 200 nm, as described elsewhere [19]. LUVs, loaded with a NaCl aqueous solution (451 mM and 20 mM phosphate buffer, pH 7.2), were extruded through a polycarbonate membrane with a LiposoFast basic extruder (Avestin Inc., Ottawa, ON, Canada) and dialyzed against a sodium sulphate (150 mM and 20 mM phosphate buffer, pH 7.2) aqueous solution, to remove un-encapsulated chloride. The chloride release was monitored employing a chloride-selective electrode (Hach 9652C, Loveland, CO, USA) and the lipid concentration during the experiments was 0.5 mM. In a typical experiment, the sodium bicarbonate aqueous solution, prepared with the sodium sulphate one, was added at t = -10 s; the bicarbonate concentration during the experiments was 40 mM. At t = 0 s a solution of the corresponding compound in DMSO was added; at t = 300 s a surfactant (Triton-X, 10% dispersion in water, 20 µL) was added to lyse the vesicles and release all the encapsulated chloride. This value was considered as 100% release and used as such. EC_50_ values were calculated from plotting the normalized chloride efflux observed for each anionophore concentration at 300 s against such concentrations and fitting the resulting curve by using Hill analysis:(1)$$\%efflux=V_{max}\frac{{\left[compound\right]}^{n}}{{EC_{50}}^{n}+{\left[compound\right]}^{n}}$$
where *V_max_* is the maximum chloride efflux, *EC_50_* the concentration of compound needed to induce a 50% chloride release at 300 seconds and *n* the Hill parameter.

### 4.3. Halide influx Assay on FRT Cells

FRT cells were stably transfected with a halide-sensitive yellow fluorescent protein (YFP-H148Q/I152L) and cultured as previously described [37,38]. The activity of the anionophores was determined in FRT cells using a fluorescence plate reader (Tristar2 S, Berthold Technologies, Bad Wildbad, Germany) equipped with 485 nm excitation and 535 nm emission filters. Functional experiments were carried out 48 h after cell seeding and 30 min before the assay the cells were washed twice with a solution containing (in mM): NaCl 136, KNO_3_ 4.5, Ca(NO_3_)_2_ 1.2, MgSO_4_ 0.2, glucose 5, HEPES 20 (pH 7.4). The cells were incubated in 60 μL of this solution at 37 °C with the anionophores or with DMSO as control. Once the assay had started, the fluorescence was recorded every 0.2 s for between 65 s for each well. At 5 s after the start of fluorescence recording, 100 μL of an extracellular solution containing 136 mM NaI instead of NaCl, were injected so that the final concentration of NaI in the well was 85 mM. The iodide influx was assessed as a quenching of fluorescence, as the anion binds to the intracellular YFP. The initial rate of fluorescence decay (QR) was derived by fitting the signal to a double exponential function, after background subtraction and normalization for the average fluorescence before NaI addition. The QR is a direct indication of the halide transport activity of the tested compound [39].

### 4.4. HBEC Culture

The HBEC isolation, culture and differentiation methods were previously described in detail [23]. Briefly, HBEC were selected from mainstem human bronchi, derived from CF and non-CF individuals undergoing lung transplant. Cells were obtained from two non-CF donors (patients with pulmonary hypertension and emphysema) and two CF patients with F508del/F508del genotype. The collection of bronchial epithelial cells was specifically approved by the Ethics Committee of the Istituto Giannina Gaslini following the guidelines of the Italian Ministry of Health (updated registration number: ANTECER, 042-09/07/2018). Each patient provided informed consent to the study using a form that was also approved by the Ethics Committee.

Cells were detached by overnight incubation of bronchi at 4 °C in a solution containing protease XIV and then cultured in a serum-free proliferative medium as described before. We demonstrated that this medium favours cell number amplification [40].

To obtain differentiated epithelia, HBEC were seeded at high density on Snapwell permeable supports (Corning, code 3801, New York, NY, USA) with bilateral addition of a serum-free medium (LHC9-RPMI). After 48 h the medium was replaced only on the basolateral side (Air-Liquid-Interface condition, ALI) with DMEM/F12 (1:1) plus 2% New Zealand fetal bovine serum (Life Technologies, Monza, Italy), hormones and supplements.

The cells were maintained in ALI condition for 4–5 weeks before performing the experiments. The complete epithelia differentiation was checked by measuring the transepithelial electrical resistance (TEER) and potential difference with an epithelial voltohmmeter EVOM2 (World precision Instrument, Sarasota, FL, USA).

### 4.5. Cell Viability

Cell toxicity of the anionophores was evaluated in non-CF HBEC by the Trypan blue exclusion assay [41]. Cells were exposed to MM3 and MM34 for 24 h and the concentrations tested were (in μM): 4, 2, 1, 0.8, 0.4, 0.2, 0.1, 0.05 and 0 (vehicle, DMSO). A number of five replicates for each condition was performed. After exposure, cells were trypsinized and harvested for toxicity evaluation. To avoid an under-estimation of the amount of dead cells, also the cells that had been detached from the plate during the anionophore’s exposition were collected and assayed. In each experiment, an amount of at least 450 cells was considered. The concentration to have the median toxic dose, TD_50_, was calculated plotting the percentage of cell survival against the anionophore concentration (A) and fitting the data with:(2)$$\%survival=100(1-\frac{A}{\left(TD_{50}+A\right)}[1-Tox_{max}])$$
where *Tox_max_* is the maximum toxicity induced by the anionophore.

### 4.6. Trans-Epithelial Electrical Conductance (TEEC) Measurement

CF epithelial monolayers were treated for 24 h with 130 µL of a modified buffer containing (in mM): 150 NaCl, 5 KCl, 1.2 CaCl_2_, 0.5 MgCl_2_ and 0.1 HEPES (pH 7.4) and 0.25 µM of compound or 0.1% DMSO as vehicle. TEER measurements were performed using the EVOM2 system and the resistance values were taken at resting condition and after incubation. The measurements were then converted into their reciprocal conductances.

### 4.7. Fluid re-Absorption Measurement

The apical surface of differentiated epithelia was treated with 130 µL of a modified buffer and 0.25 µM of compound or 0.1% DMSO as vehicle. After 24 h, the remaining mixture of fluid and mucus was carefully recovered and weighted. The net flux *J_W_* across the epithelium was calculated as:(3)$$Jw=\frac{V_{i}-V_{f}}{At}$$
where *V_i_* and *V_f_* are the initial and final apical volumes, *A* is the epithelium area (for Snapwell support: 1.13 cm^2^) and t is the time interval between addition of *V_i_* and recovering of the remaining fluid *V_f_*.

### 4.8. Micro-Rheology

The fluid recovered from the apical side of epithelia was also analysed for the micro-rheology properties using the Multiple Particle Tracking (MPT) as described [42,43,44,45]. If not immediately tested, the samples may be frozen and stored at −80 °C up to three weeks. In MPT, the time course of the position of nano-spheres in suspensions inside the medium to be studied is recorded. An aliquot of 25 µL of mucus and 1 µL of solution containing polysterene carboxylated yellow/green fluorescent beads of 200 nm diameter (λ_exc_ = 488 nm, λ_em_ = 505–515 nm; Fluospheres, Life Technologies) were mixed, and a sample of 8 µL of mucus containing <100 beads/field was deposited between two glass coverslips, and the borders were sealed to avoid evaporation. After an equilibration at room temperature for 10 min, beads positions were focused to the mid-height of the sample to exclude beads that might be interacting with the coverslips with a 60× (N.A. = 1.42) oil immersion objective connected to a CCD videocamera. Images were captured at a rate of 6 frames, of 1392 × 1040 pixels, per second. The trajectory of the beads was recorded using the Multitracker plug-in of ImageJ [43]. About 400 beads were tracked in four to eight fields per sample.

The movement of the fluorescent beads in a given time interval, τ, is described by its mean squared displacement, <msd>; the displacement of beads in viscous solutions displays a linear dependence with respect to the time interval τ of the form:(4)$$<msd(\tau )>=4D_{0}\tau$$
from which it is possible to calculate *D_o_*, the diffusion coefficient. The viscosity η was calculated from the Stokes–Einstein equation, as:(5)$$\eta =\frac{k_{B}T}{6\pi D_{0}r}$$
where *k_B_* is the Boltzmann constant, *T* is the absolute temperature and *r* is the radius of the microsphere.

### 4.9. Mucus pH Measurement

To verify the impact of the anionophores on the ASL pH the epithelia-secreted mucus was treated as described above, quickly collected from the apical surface and immediately measured using a microelectrode (SevenCompact-Mettler Toledo, Novate Milanese, Italy).

### 4.10. Statistics

All statistical analysis and micro-rheology calculations were done with IgorPro 8.04 software (Lake Oswego, Oregon, USA). Data are shown as mean ± SEM. Comparison between a group against a specified value was done using the non-paired Student’s *t* test. For conductance experiments, the paired Student’s *t*-test was applied. A value of *p* < 0.05 was considered statistically significant (* means *p* < 0.05).

## 5. Data Availability

All data generated and analysed during this study are included in this article. All data is available from the corresponding authors upon reasonable request.

## Figures and Tables

**Figure 1 ijms-21-01488-f001:**
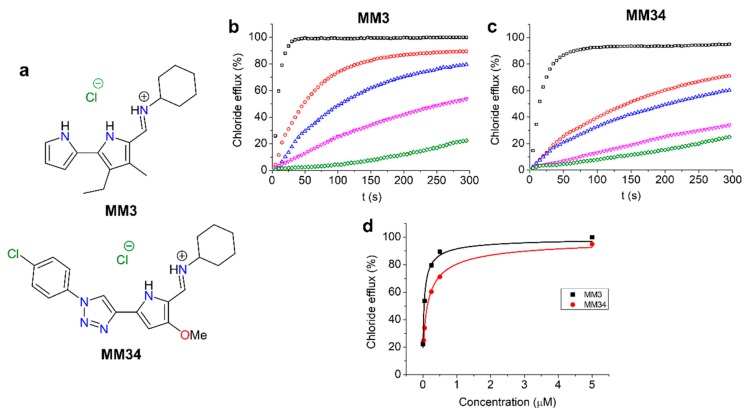
(**a**) Structures of the studied compounds. (**b**) Chloride efflux promoted by compound MM3 (5 µM, black; 0.5 µM, red; 0.25 µM, blue; 0.05 µM, magenta; 0.005 µM, green) in unilamellar 1-palmitoyl-2-oleoyl-glycero-3-phosphocholine (POPC) vesicles. Vesicles containing NaCl (451 mM NaCl and 20 mM phosphate buffer, pH 7.2) were immersed in Na_2_SO_4_ (150 mM Na_2_SO_4_, 40 mM NaHCO_3_ and 20 mM phosphate buffer, pH 7.2). Each trace represents the average of at least three different experiments. (**c**) Chloride efflux promoted by compound MM34 (5 µM, black; 0.5 µM, red; 0.25 µM, blue; 0.05 µM, magenta; 0.025 µM, green) in unilamellar POPC vesicles. The experimental conditions are identical to those reported for MM3. (**d**) Transport activity plotted againts the concentration for both compounds, in model liposomes, was fitted with the Hill equation, represented as a continuous line.

**Figure 2 ijms-21-01488-f002:**
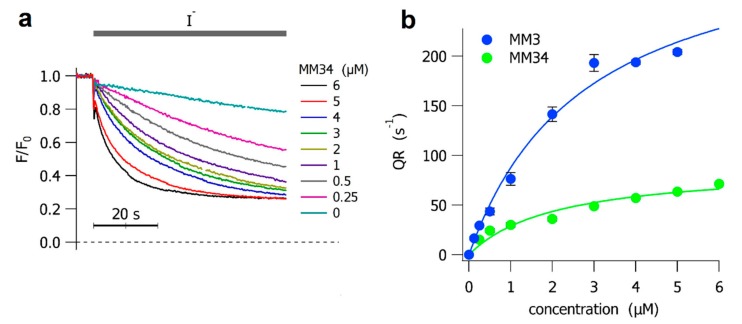
Anionophore-driven halide transport in Fischer Rat Thyroid (FRT) cells. (**a**) The time course of the fluorescence decay in FRT cells incubated at different concentrations of MM34, as indicated. The addition of iodide is indicated by the gray line over the traces. Cells treated with DMSO were used as control. The time-calibration bar is shown below the traces. (**b**) The initial YFP-quenching rate (QR) obtained from FRT cells treated with different concentrations of MM3 (blue) and MM34 (green). Each point represents the means ± SEM from 8 experiments. The concentration–response curves, fitted with a Langmuir model, are represented as continuous lines.

**Figure 3 ijms-21-01488-f003:**
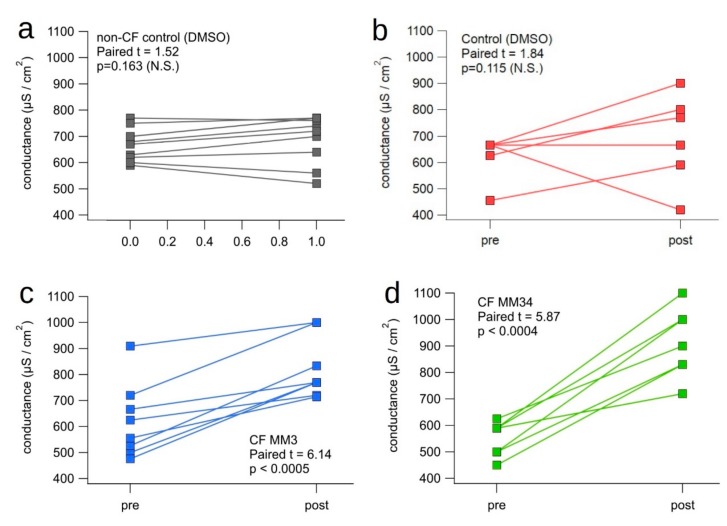
Transepithelial electrical conductance (TEEC) modified by the anionophores-driven transport in CF-HBEC. (**a**) HBEC monolayers derived from 2 non-CF donors treated with 0.1% DMSO are presented as control (*n*= 10). HBEC monolayers derived from 2 homozygous F508del-CFTR patients, treated for 24 h with (**b**): 0.1% DMSO (*n*= 6); (**c**): 0.25 µM of MM3; (**d**): 0.25 µM of MM34 (*n* = 9). Conductance was measured before (pre) and after treatment (post). The Student’s paired-*t* test parameter and the corresponding *p* value are indicated in each panel.

**Figure 4 ijms-21-01488-f004:**
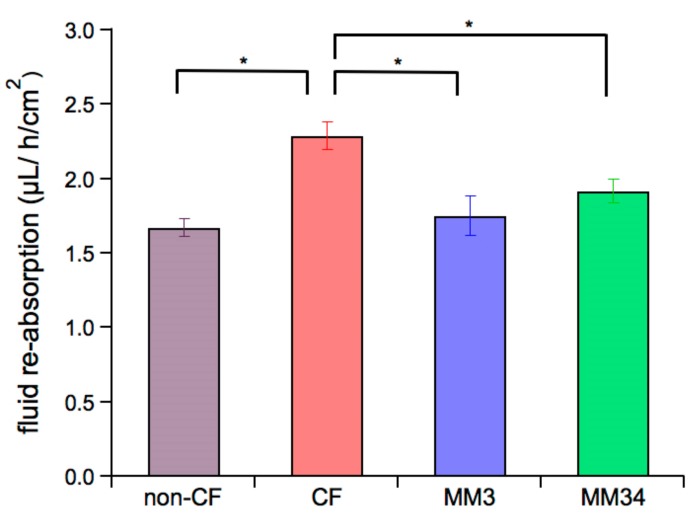
Impact of the anionophores on net fluid re-absorption in CF epithelia. Bars indicate the fluid re-absorption rate measured after 24 h of incubation. Untreated CF (*n* = 7) and non-CF (*n* = 9) epithelia were incubated with 0.1% DMSO as control. CF epithelia were incubated with 0.25 µM of MM3 (*n* = 7) or 0.25 µM of MM34 (*n* = 7). Data were obtained from two non-CF donors and two different homozygous F508del-CFTR patients. Mean values ± SEM are shown. The asterisk (*) indicates a significant difference (*p* < 0.05) obtained from a Student’s non-paired-*t* test.

**Figure 5 ijms-21-01488-f005:**
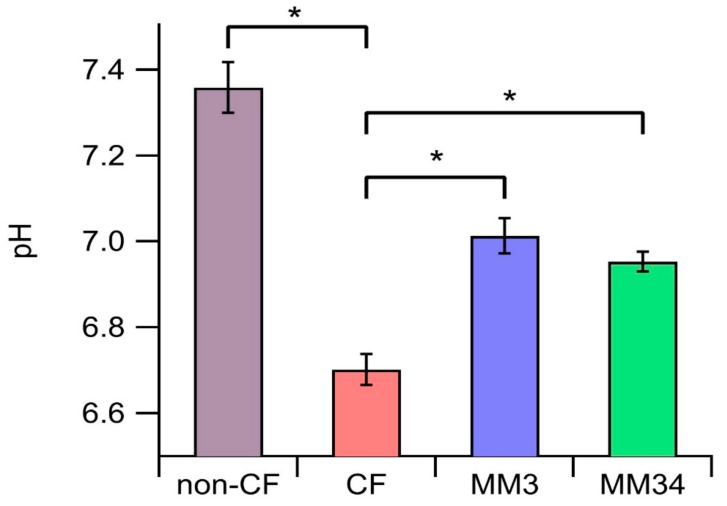
Evaluation of the apical fluid pH. Bars indicate the average pH after 24 h of incubation. Untreated CF (*n* = 17) and non-CF (*n* = 8) epithelia were incubated with 0.1% DMSO as control. CF epithelia were incubated with 0.25 µM of MM3 (*n* = 9) or 0.25 µM of MM34 (*n* = 9). Data were obtained from two non-CF donors and two different homozygous F508del-CFTR patients. The asterisk (*) indicates a significant difference (*p* < 0.05) obtained from a Student’s non-paired-*t* test.

**Figure 6 ijms-21-01488-f006:**
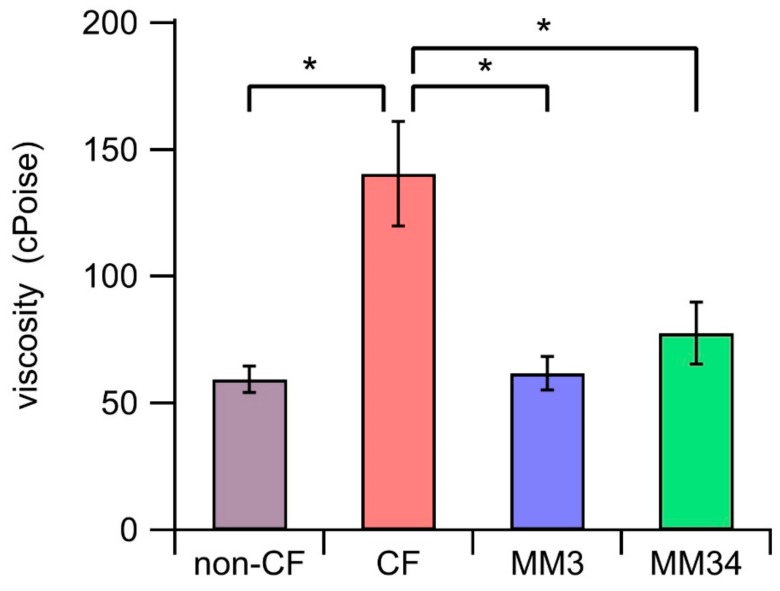
Mucus viscosity. Bars represent the average viscosity after 24 h of incubation. Untreated CF (*n* = 8) and non-CF (*n* = 6) epithelia were incubated with 0.1% DMSO as control. CF epithelia were incubated with 0.25 µM of MM3 (*n* = 7) or 0.25 µM of MM34 (*n* = 8). Data were obtained from two non-CF donors and two different homozygous F508del-CFTR patients. The asterisk (*) indicates a significant difference (*p* < 0.05) obtained from a Student’s non-paired-*t* test.

**Table 1 ijms-21-01488-t001:** Transepithelial electrical resistance (TEER) measured in Cystic fibrosis (CF)-human bronchial epithelial cells (HBEC) before and after the treatment with the anionophore.

	DMSO		MM3		MM34	
	pre	post	pre	post	pre	post
**Average ± sem (Ω)**	1400 ± 94	1500 ± 115	1370 ± 158	1840 ± 124	1250 ± 63	1980 ± 146

**Table 2 ijms-21-01488-t002:** Differences of the conductance (in µS/cm^2^) measured before and after the treatment with the anionophores. The difference of the average of each group against 0 was determined with the Student’s *t* test; difference is considered significant for *p* < 0.05.

	DMSO	MM3	MM34
**average**	66.7	207.3	326.7
**sem**	70.3	31.1	46.6
***n***	6	9	9
***p***	0.386 (N.S.)	0.0002	0.0001

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
