# Peer review of "Small Molecule Anion Carriers Correct Abnormal Airway Surface Liquid Properties in Cystic Fibrosis Airway Epithelia"

_ijms, 2020, doi:10.3390/ijms21041488_

Round 1

Reviewer 1 Report

The strategy of using anionophores to substitute the defective CFTR protein is getting quite a lot of attention as it seems to be advantageous over existing clinically available CFTR modulators and independent of the genetic background of CF patients. Indeed, recent investigations by other groups (Li et al, 2019) and the same group (Fiore et al, 2018, Hernando et al, 2018) were performed in physiologically less relevant cell models (FRT cells). The primary goal of the current work is to assess the effect of two anionophores MM3 and MM34 in HBE cells and assessed parameters such as Transepithelial conductance and ASL properties. Although the study is interesting and presenting new information about two anionophores MM3 and MM34, there are some important issues that need to be addressed, as indicated in my specific comments.

Specific comments:

1)    Was there any change in Ciliary beating frequency (CBF) with the usage of anionophores in HBE ALI cultures?  As it is evident from the Fig. 6 that there was a statistically significant reduction in the ASL mucus viscosity.

2)    In Fig. 2, authors have assessed transepithelial conductance in CF epithelia but not in non-CF epithelia? It would be useful if authors could provide TEER values (pre and post) separately (to ensure that these compounds are not affecting monolayer integrity).

3)    The authors need to mention about t-test parameter (Paired or unpaired observation) in Fig. 4, 5 and 6.

4)    Comparative aspect with clinically given CFTR modulators (VX-809, VX-661, and VX-770) is missing in figure 4,5, and 6 to show that MM3 and MM34 are better than corrector molecules.

5)    A recent article from the same group, Fiore et al, 2018, has demonstrated the efficacy of MM3, not MM34 in FRT cells expressing wt, G551D, and F508del-CFTR with and without CFTR modulators, does MM34 have same pharmacological profile as that of MM3? In more detail, it would be interesting if authors could perform a similar set of experiments in HBE cells with a diverse genetic background (or Authors could perform experiments in FRT cells expressing nonsense mutations and then validate (confirm) the findings in HBE cells), mainly emphasizing on nonsense mutations.

6)    It would add more value to the study if authors perform Trans epithelial current measurements in HBE or HNE’s to further demonstrate the efficacy of MM3 and MM34 and check important electrophysiological parameters such as Amiloride inhibited, CFTR-Inh172 inhibited, and Fsk, ATP/UTP (TMEM16A)  activated short circuit currents. Also, it would give information if these molecules are affecting activation of CFTR or interfering with potentiator or corrector (For e.g. as shown in Fiore et al, 2018, BJP Page 1774 in FRT cells).

7)    The authors need to increase the number of CF donors to a minimum of 4 - 5 donors to make the findings sound and more convincing. These suggestions (points 1-7) do not change the interpretations or validity of findings but provide experimental clarity and strengthen the findings/claims made by the Authors.

Author Response

Responses to Reviewer 1 Comments

We would like to thank the reviewer for the appreciation and the provided comments. We provide below a point by point response and we believe that the implemented changes have improved the manuscript. Unfortuntely, we could not perform all of them due to limited period for revise this manuscript. We hope in the future to be able to complete our work.

Point 1: Was there any change in Ciliary beating frequency (CBF) with the usage of anionophores in HBE ALI cultures?  As it is evident from the Fig. 6 that there was a statistically significant reduction in the ASL mucus viscosity.

Response 1: This is a very interesting suggestion. Indeed, the reduction of the ASL mucus viscosity would facilitate the ciliary beating. In fact, we have planned to measure the ciliary movements in the continuation of this project.

Point 2: In Fig. 2, authors have assessed transepithelial conductance in CF epithelia but not in non-CF epithelia? It would be useful if authors could provide TEER values (pre and post) separately (to ensure that these compounds are not affecting monolayer integrity).

Response 2: This is a very improving suggestion. Non-CF conductance data are now included in the manuscript.

Point 3: The authors need to mention about t-test parameter (Paired or unpaired observation) in Fig. 4, 5 and 6.

Response 3: We have modified the caption to the figures to explicitly mention whether the Student's t-test was paired or unpaired.

Point 4: Comparative aspect with clinically given CFTR modulators (VX-809, VX-661, and VX-770) is missing in figure 4,5, and 6 to show that MM3 and MM34 are better than corrector molecules.

Response 4: The effect of the VX-809 on the ALI was already shown previously by Gianotti et al. (J. Cystic Fibr,  2016, 15, 295-301) Corrector VX-809 reduces the viscosity but not the fluid absorption in CF epithelia. This is, perhaps, correlated with the increase of bicarbonate permeability of the VX-809-corrected CFTR  (Ferrera et al., J. Cystic Fibr. 2019, 18, 602–5). A comment is now included in the discussion.

Point 5: A recent article from the same group, Fiore et al, 2018, has demonstrated the efficacy of MM3, not MM34 in FRT cells expressing wt, G551D, and F508del-CFTR with and without CFTR modulators, does MM34 have same pharmacological profile as that of MM3? In more detail, it would be interesting if authors could perform a similar set of experiments in HBE cells with a diverse genetic background (or Authors could perform experiments in FRT cells expressing nonsense mutations and then validate (confirm) the findings in HBE cells), mainly emphasizing on nonsense mutations.

Response 5:  We agree with the reviewer that a more extensive mutation benchmark would be optimal to emphasize the potential of anionophores in CF therapy. We are, indeed, engaged in these kind of experiments, that expect to complete in the next months. In this article we intend to present a proof of concept, analysing the most common CF mutation. Regarding the compound MM34, it was synthetized before we published the mentioned article (Fiore, 2019), and its pharmacological characterization is still in course. 

Point 6: It would add more value to the study if authors perform Trans epithelial current measurements in HBE or HNE’s to further demonstrate the efficacy of MM3 and MM34 and check important electrophysiological parameters such as Amiloride inhibited, CFTR-Inh172 inhibited, and Fsk, ATP/UTP (TMEM16A)  activated short circuit currents. Also, it would give information if these molecules are affecting activation of CFTR or interfering with potentiator or corrector (For e.g. as shown in Fiore et al, 2018, BJP Page 1774 in FRT cells).

Response 6: Thanks for this improving suggestion, unfortunately, due to the limited time for revise this manuscript we could not perform the requested experiments. However, we will take in consideration this valuable comment for the continuation of this project.

Point 7: The authors need to increase the number of CF donors to a minimum of 4 - 5 donors to make the findings sound and more convincing. These suggestions (points 1-7) do not change the interpretations or validity of findings but provide experimental clarity and strengthen the findings/claims made by the Authors.

Response 7: Unfortunately, we have completed these experiments with only two donors. Because the limited time for revise this manuscript conceals the possibility to increase the number of subjects.

Reviewer 2 Report

The authors present data on the effects of anionophores on the key properties of the airway surface liquid (ASL) in epithelial cell models of bronchial epithelial cells, including primary differentiated human bronchial epithelial cells (HBEC). The significance of the work is related to the potentially beneficial effect of the anionophores to improving the properties of ASL in patients with cystic fibrosis independent of the CFTR gene mutation.  

The major concern is the high concentration of vehicle control used in the study 0.1% DMSO.

The other major concern is that the authors do not show the effects of the DMSO control, along with the effects of the two anionophores.

In view of the above concerns, the clinical significance of the finings described by the authors is not convincing.

Major:

Figure 3a. The effects of DMSO control: It is concerning that 4 out of 6 monolayers experienced increased conductance in the presence of 0.1% DMSO in range similar to MM3 or MM34 treatment. The difference from the later treatments, shown in Figures 3b and 3c, is that all monolayers (7/7 and 8/8, respectively) showed increased conductance. The 0.1% DMSO used as control is concerning. There are many non-specific effects of DMSO in cultured cells.

Figure 4. Please, provide control data showing effects of 0.1% DMSO on fluid reabsorption in F508del homozygous HBEC to demonstrate specificity on the effects by MM3 or MM34. Please, specify the disease background of the non-CF HBEC used in the study.

Figure 5. Please, provide control data showing effects of 0.1% DMSO on pH in F508del homozygous HBEC to demonstrate specificity on the effects by MM3 or MM34. Please, specify the disease background of the non-CF HBEC used in the study.

Figure 6. Please, provide control data showing effects of 0.1% DMSO on mucus viscosity in F508del homozygous HBEC to demonstrate specificity on the effects by MM3 or MM34. Please, specify the disease background of the non-CF HBEC used in the study.

The Figure 6 legend should be changed to indicate mucus viscosity instead of ASL mucus viscosity. The mucus is separate from ASL.  

Minor:

The authors should improve the English language before consideration for publication.

Below are just few examples.

Title: use lower case for words inside the title

Abstract:

24: “The reduced and acidic ASL” needs clarification. Please, clarify the phrase.

29: “Human primary bronchial epithelial cells …_ should be “Primary human…_

30: What are “mucociliated epithelia”?

32: “periciliary” should be “periciliary”

33: “pH value”: remove the work “value”

55: error, illegible run-in words, unable to evaluate the text

85; explain abbreviation “LUVs”

Author Response

Response to Reviewer 2 Comments

We would thank the reviewer for the consideration and useful comments. We understand the concerns about DMSO effects and we provide below a point by point response to all the raised issues. We hope to have adequately presented evidences about the safety of this dose.

Point 1:  The major concern is the high concentration of vehicle control used in the study 0.1% DMSO.

Response 1: It is a normal practice in our laboratories, as well in other's labs, to use a DMSO as a drug vehicle at concentrations ≤0.1%. In our experience, we have not observed any major effect of DMSO at such low concentration in most of the parameters measured, either in isolated cells or in cell monolayers. To reinforce this concept, we have now included a specific reference where the effect of DMSO in bronchial epithelial cells is discussed (Tomati et al, JCI Insight. 2018).

Point 2: The other major concern is that the authors do not show the effects of the DMSO control, along with the effects of the two anionophores.

Response 2: We have indeed checked the effect of the DMSO vehicle alone. For correctness, we have compared the data from not treated epithelia with DMSO treated epithelia and we did not found any difference.

Point 3: In view of the above concerns, the clinical significance of the finings described by the authors is not convincing.

Response 3: The purpose of this article is to present a proof of concept regarding the possibility of modifying key ASL parameters in a model bronchial epithelia by the anionophore treatment.

It is possibly true that there is still a long way before data with convincing clinical significance can be obtained. However, this work could be considered a promising start for a new paradigm in the research of CF therapy, using synthetic compound that could substitute the damaged or absent CFTR.

Point 4: Figure 3a. The effects of DMSO control: It is concerning that 4 out of 6 monolayers experienced increased conductance in the presence of 0.1% DMSO in range similar to MM3 or MM34 treatment. The difference from the later treatments, shown in Figures 3b and 3c, is that all monolayers (7/7 and 8/8, respectively) showed increased conductance. The 0.1% DMSO used as control is concerning. There are many non-specific effects of DMSO in cultured cells.

Response 4: Unfortunately, we deal with a biological system that is characterized by variabilities. Thus, our conclusions are not based on single results, but in the average of various experiments in a given condition. As we point out along the text, it has to be observed that, in average, there is any statistical change in the conductance of epithelia treated with DMSO, but a significant difference when the preparation is treated with DMSO plus an anionophore. We must conclude that the difference is due to the anionophore and not to DMSO.

Point 5-6-7: Figure 4. Please, provide control data showing effects of 0.1% DMSO on fluid reabsorption in F508del homozygous HBEC to demonstrate specificity on the effects by MM3 or MM34. Please, specify the disease background of the non-CF HBEC used in the study.

Figure 5. Please, provide control data showing effects of 0.1% DMSO on pH in F508del homozygous HBEC to demonstrate specificity on the effects by MM3 or MM34. Please, specify the disease background of the non-CF HBEC used in the study.

Figure 6. Please, provide control data showing effects of 0.1% DMSO on mucus viscosity in F508del homozygous HBEC to demonstrate specificity on the effects by MM3 or MM34. Please, specify the disease background of the non-CF HBEC used in the study.

Response 5-6-7: As mentioned along the text, all controls were treated with the anionophore vehicle, DMSO 0.1%. Indeed, in all figures  “CF” and “non-CF” are explicitly referred to epithelia treated with DMSO 0,1%  as control experiments in the captions.

The disease background of the non-CF are cited in Material and Methods-HBEC culture.

Point 8: The Figure 6 legend should be changed to indicate mucus viscosity instead of ASL mucus viscosity. The mucus is separate from ASL.  

Response 8: We have modified as suggested by the reviewer.

Point 9:  The authors should improve the English language before consideration for publication.

Response 9: We have now revised the manuscript at our best.

Point 10:  Title: use lower case for words inside the title

Point 11:  “The reduced and acidic ASL” needs clarification. Please, clarify the phrase.

Point 12:  “Human primary bronchial epithelial cells …_ should be “Primary human…_

Point 14: “periciliary” should be “periciliar”

Point 15: “pH value”: remove the work “value”

Response 8-15:  Thanks for these suggestions, all revised in manuscript.

Point 13: What are “mucociliated epithelia”?

Response 13:  It refeers to a ciliated columnar epithelium, typical of the airways mucose,  with ciliate cells and globet cells that produce mucine. As described in the methods, it is obtained by a protocol described by Scudieri et al. (J.Physiol, 2012)

Point 16:  Explain abbreviation “LUVs”

Response 16:  LUVs means Large Unilamelar Vesicle, as defined in the subtittle 2.1.

Round 2

Reviewer 1 Report

The manuscript is in the bits and pieces now and needs the additional suggested experiments to make it complete. Authors could improve the manuscript with the addition of more experimental data. Indeed ALI cultures and ion trasnport studies takes time so authors should focus on getting quality data out of it with more time in hand and improve the quality of the manuscript for reconsideration.  

Author Response

We understand your concerns but we could not perform all the requested experiments due to limited period for revise this manuscript

Reviewer 2 Report

The authors addressed my concerns and comments.

I would like to suggest replacing the word "comparison" with "control" in the legend for Figure 3.

Author Response

Thanks for your suggestion, We have replaced the word.